# Inhibitory Mechanism of *Trichoderma virens* ZT05 on *Rhizoctonia solani*

**DOI:** 10.3390/plants9070912

**Published:** 2020-07-19

**Authors:** Saiyaremu Halifu, Xun Deng, Xiaoshuang Song, Ruiqing Song, Xu Liang

**Affiliations:** 1College of Forestry, Northeast Forestry University, Harbin 150040, China; saiyaremu@nefu.edu.cn; 2Institute of Forestry Protection, Heilongjiang Forestry Academy, Harbin 150040, China; dxhappy@126.com (X.D.); sxshappy@126.com (X.S.); 3Beijing DaXing District Forestry Workstation, No. 17 Administrative Street, Huangcun Town, Beijing 102600, China; xuliang19824@163.com

**Keywords:** *Trichoderma virens* ZT05, *Rhizoctonia solani*, antagonism, antifungal effect of metabolites, transcriptome sequencing analysis

## Abstract

*Trichoderma* is a filamentous fungus that is widely distributed in nature. As a biological control agent of agricultural pests, *Trichoderma* species have been widely studied in recent years. This study aimed to understand the inhibitory mechanism of *Trichoderma virens* ZT05 on *Rhizoctonia solani* through the side-by-side culture of *T. virens* ZT05 and *R. solani*. To this end, we investigated the effect of volatile and nonvolatile metabolites of *T. virens* ZT05 on the mycelium growth and enzyme activity of *R. solani* and analyzed transcriptome data collected from side-by-side culture. *T. virens* ZT05 has a significant antagonistic effect against *R. solani*. The mycelium of *T. virens* ZT05 spirally wraps around and penetrates the mycelium of *R. solani* and inhibits the growth of *R. solani*. The volatile and nonvolatile metabolites of *T. virens* ZT05 have significant inhibitory effects on the growth of *R. solani*. The nonvolatile metabolites of *T. virens* ZT05 significantly affect the mycelium proteins of *R. solani*, including catalase (CAT), superoxide dismutase (SOD), peroxidase (POD), selenium-dependent glutathione peroxidase (GSH-Px), soluble proteins, and malondialdehyde (MDA). Twenty genes associated with hyperparasitism, including extracellular proteases, oligopeptide transporters, G-protein coupled receptors (GPCRs), chitinases, glucanases, and proteases were found to be upregulated during the antagonistic process between *T. virens* ZT05 and *R. solani*. Thirty genes related to antibiosis function, including tetracycline resistance proteins, reductases, the heat shock response, the oxidative stress response, ATP-binding cassette (ABC) efflux transporters, and multidrug resistance transporters, were found to be upregulated during the side-by-side culture of *T. virens* ZT05 and *R. solani*. *T. virens* ZT05 has a significant inhibitory effect on *R. solani*, and its mechanism of action is associated with hyperparasitism and antibiosis.

## 1. Introduction

The damping-off disease of Mongolian pine (*Pinus sylvestris var. mongolica*) is affected by pathogenic fungi, including *Fusarium oxysporum*, *Rhizoctonia solani* J.G. Kühn, and *Pythium*, among which *R. solani* J.G. Kühn is the major pathogenic factor. The damping-off disease of *Pinus sylvestris var. mongolica* mostly occurs during years 1–3 of seedling caused by *R. solani* J.G. Kühn infection, resulting in a death rate of 60% or no production [1,2,3]. Chemical controls have been applied during nursery to reduce fungal infections because of their effectiveness, convenience, and economic benefits. However, the overuse of chemicals disrupts the soil microbial balance leading to chemical resistance in pathogens and environmental pollution [4,5,6]. Biological controls use beneficial microorganisms and microbial metabolites to prevent plant diseases. Mechanisms of biological controls in the inhibition of pathogen growth include microbial competition, antibiosis, hyperparasitism, cross-protection, and induced resistance. Biological controls are beneficial to the environment with no pollution or residue and avoid killing natural enemies of pathogens. Biological controls reduce drug resistance in pathogens and are conducive to human and animal safety. Biological controls not only protect plant growth but also effectively treat plant diseases resulting in increased growth and harvest [7,8,9]. Biological controls focus on the virtuous cycle of the ecosystem and protect the environment through the application of biofungi, including *Trichoderma*, *Chaetomium*, *yeast strains*, *Paecilomyces lilacinus*, *Verticillium chlamydosporium*, and mycorrhizal fungi. The inhibitory effect of *Trichoderma* species on plant pathogens has been widely studied and applied [10,11,12,13].

*Trichoderma* species belong to *Deuteromycotina*, *Hyphomycetes*, and *Hyphomyceteales*. They have wide adaptability and survivability and are widespread [14]. The biological control mechanisms of Trichoderma species include competition for limited nutrients and living space [15,16], hyperparasite infection on plant pathogens [17], inhibitory effect on the growth of plant pathogens [18,19], improvement of plant growth, and the induction of plant resistance to pathogens [20]. *Trichoderma* species not only grow and reproduce rapidly but also use limited nutrients and rapidly adapt to limited living space. They advantageously inhibit the growth of pathogens via interaction with pathogenic fungi. Previously, Howell et al. [21] showed that *Trichoderma* occupied culture space quickly under suitable growth conditions and gradually covered the surface of the agar plate to inhibit the growth of the pathogenic fungi *R. solani*. One of the important methods used by *Trichoderma* species to kill pathogens is through hyperparasite infections [22,23,24]. During hyperparasite infection, gene expression of proteases associated with host recognition, oligopeptide transporters, and GPCRs was significantly upregulated, which played a vital role in the identification of pathogens and hyperparasite signaling [25,26,27]. The expression of genes related to hydrolase secretion was significantly upregulated during the hyperparasite infection of *Trichoderma* within hosts. By knocking out these genes, the hyperparasitic ability of *Trichoderma,* within hosts, was reduced [28,29,30,31]. Barbara et al. showed that the knockout of *T. virens* gene *tgaA* on the G protein alpha subunit reduced the hyperparasitic ability of *Trichoderma* within hosts, while the expression of chitinase genes *ech42* and *nag1* was significantly reduced [32]. In addition, the knockout of the MPAK gene reduced the hyperparasitic ability of *T. atroviride* [33]. Antibiosis is the main mechanism of *Trichoderma* species in biological controls. During the antagonistic interaction between *Trichoderma* and its hosts, *Trichoderma* secretes volatile and nonvolatile metabolites to inhibit the growth of pathogenic microorganisms [34,35]. Additionally, *Trichoderma* degrades toxic substances secreted by pathogenic microorganisms through ABC transporter proteins, heat shock proteins, etc., thereby reducing the inhibitory effect of pathogenic microorganisms on *Trichoderma* [36,37,38,39].

Transcriptome, also known as “expression profile”, refers to all mRNAs that are involved in protein translation [40,41,42]. RNA-seq is the second generation high-throughput transcriptome sequencing, which has been widely used in recent years. RNA-seq allows high sequencing throughput and covers a wide detection range. It is highly sensitive and can detect new genes. RNA-seq identifies genes that are significantly differentially expressed between samples. The main biological functions and metabolic pathways of these differentially expressed genes are speculated through annotation and gene enrichment analysis. Key functional genes are also identified during gene analysis [43,44,45,46]. Transcriptomics has become an effective method to investigate the biological control of *Trichoderma*, along with sequencing multiple *Trichoderma* genomes [47,48]. The study of the molecular mechanism of *Trichoderma* for inhibiting pathogenic organisms helps to improve the biological control efficiency of *Trichoderma* and to reduce the economic losses in agriculture and forestry [49,50]. In this paper, *T. virens* ZT05 strain was studied, which was isolated from the rhizosphere soil of *Pinus sylvestris var. mongolica* in Zhanggutai (Liaoning Province, China). *T. virens* ZT05 was cultured side-by-side with *R. solani* and observed under electron microscopy. The effect of both volatile and nonvolatile metabolites from *Trichoderma* on the mycelium growth and enzyme activity of *R. solani* was investigated. The inhibitory effect of *T. virens* ZT05 on *R. solani* was discussed based on the results of transcriptome sequencing analysis of the side-by-side cultured *T. virens* ZT05 and *R. solani*.

## 2. Materials and Methods

### 2.1. Strain and Culture Conditions

*Trichoderma virens* ZT05 was isolated from the rhizosphere soil of the *P. sylvestris var. mongolica* forest of the Zhanggutai Experimental Forest Farm of Liaoning Province (42°43′–42°51″ N, 121°53′–122°22″ E), China. *Rhizoctonia solani* isolated from the rhizosphere soil of the *P. sylvestris var. mongolica* seedling nursery of the Harbin Weihai forestry bureau nursery of Heilongjiang Province (Strains *Trichoderma virens* ZT05, *Rhizoctonia solani* stored in the forest microbiology laboratory of college of forestry, Northeast Forestry University). These two strains were grown on a PDA medium (potato extract 12 g/L, dextrose 20 g/L, agar 14 g/L; Haibo Biotechnology, China) at pH 6.0.

### 2.2. Antagonistic Effect of T. virens ZT05 against R. solani

*T. virens* ZT05 was cultured side-by-side with *R. solani* [51,52,53]. In the experimental group, agar plugs (diameter 10 mm) of *T. virens* strain ZT05 and *R. solani* (both cultured for three days) were inoculated on a potato dextrose agar (PDA) plate at 50 mm distance from each other. In the control group, *T. virens* strain ZT05 and *R. solani* were grown alone in individual plates. The plates were incubated at 25 °C. The experiments were repeated three times. Pathogen diameters (∅) were measured every 8 h. After 48 h, the inhibitory rate and relative inhibitory rate were calculated and plotted. The antagonistic curve was evaluated, and the competition coefficient was calculated [53,54]. Inhibitory rate and relative inhibitory rate were calculated as:Inhibitory rate (%) = ((∅ control group − ∅ experimental group)/∅ control group) × 100%(1)
Relative inhibitory rate = Inhibitory rate of R. solani/Inhibitory rate of T. virens(2)

Electron microscopy (Hitachi Japan Ltd., Quantax70) was used to observe the side-by-side culture of *T. virens* ZT05 and *R. solani*. The junction area of the two pathogens was cut using a sterile punch with a diameter of 10 mm. The excess area was removed, and the junction cut was placed in a 2% glutaraldehyde solution at 4 °C for 1 h. The junction cut was washed several times with 0.1 mol/L phosphate buffer (PBS, pH 7.0). Ethanol solutions at concentrations of 30%, 50%, 70%, 80%, and 90% (*v*/*v*) were used for dehydration (30 min per concentration). The dehydrated samples were placed in a freeze dryer (Freeze Mobile 24, Vertis Company, Inc., Gardiner, NY, USA) for freeze-drying, and then placed on a sample stage for gold spray. The samples were observed using a scanning electron microscope (SEM) [55,56,57].

### 2.3. Inhibitory Effect of the Metabolites of T. virens ZT05 on R. solani

#### 2.3.1. Inhibitory Effect of Nonvolatile Metabolites of *T. virens* ZT05 on *R. solani*

For the preparation of ethyl acetate extract, *T. virens* ZT05 was inoculated onto a PDA plate for three days, and then cut using a sterile punch with a diameter of 10 mm. Three cuts were made and placed in a 200 mL PD media and shaken at 25 °C for 150 rpm. After seven days, the culture media was filtered through eight layers of gauze. The filtrate and ethyl acetate were extracted at a volume ratio of 1:3 for two days, and then the inorganic phase was removed. The organic phase was removed using a vacuum rotary evaporator. The extract was dissolved in a 1/10 volume solution mixed of the filtrate and 10% Tween 80 solution.

To measure the inhibition effect, mycelium growth inhibition was measured [58,59]. The extract of *T. virens* ZT05 from the above was added into PDA medium and cultured with shaking to a final concentration of 20%. The 20% culture mix was poured onto plates and was inoculated with *R. solani*. As a control, a PDA plate without *T. virens* ZT05 metabolite extract was used. The plates were placed at 25 °C. The diameters of the colonies were measured every 8 h using the diagonal cross method. The inhibitory effect of nonvolatile metabolites of *T. virens* on *R. solani* was calculated and plotted. Each experiment was repeated three times. Inhibition rate of pathogens was calculated as:Inhibition rate of pathogens (%) = ((Control net growth-treatment net growth)/Control net growth) × 100%(3)

#### 2.3.2. Effects of Nonvolatile Metabolites of *T. virens* ZT05 on the Stress Resistance of *R. solani*

*R. solani* was inoculated onto a PDA plate for three days, and then cut using a sterile punch with a diameter of 10 mm. Three cuts were made and placed in 200 mL PD media and shaken at 25 °C for 150 rpm. After seven days, the culture media was filtered through eight layers of gauze, and then washed with sterile water several times to remove the culture medium on the mycelia. The *R. solani* extract was placed on sterile filter paper for water absorption, and then soaked in ethyl acetate extract prepared as described above at a concentration of 20%. As a control, 20% Tween 80 solution was used without *T. virens* metabolites. The soaking process was designed for 3, 6, 9, 12, 24, and 48 h. Then, the mycelia were removed and dried using sterile filter paper. The mycelia were ground with liquid nitrogen for tissue sample preparation. The protein expression of CAT, SOD, POD, GSH-Px, MDA, and soluble proteins were detected using a Protein Detection Kit (Jiancheng, Nanjing, China).

#### 2.3.3. Inhibitory Effect of Volatile Metabolites of *T. virens* ZT05 on *R. solani*

The up-and-down culture method was used [60]. *T. virens* ZT05 was inoculated onto a PDA plate for three days, and then the coverslip of the plate was replaced by a coverslip inoculated with *R. solani*. As a control, an empty PDA plate was topped by a cover inoculated with *R. solani*. The plates were incubated at 25 °C and observed daily. After five days, the diameter of *R. solani* was measured by a diagonal cross method. The inhibitory effect of volatile metabolites of *T. virens* on *R. solani* was calculated. Each experiment was repeated three times. The inhibition rate of pathogens was calculated as:Inhibition rate of pathogens (%) = ((∅ Control colony − ∅ treatment colony)/∅ Control colony) × 100%(4)

### 2.4. Transcriptome Sequencing Analysis of the Side-by-Side Cultured T. virens and R. solani

#### 2.4.1. Mycelium Collection and Total RNA Extraction

*T. virens* and *R. solani* (cultured for three days) were cut using a sterile punch to a diameter of 10 mm and were inoculated onto a PDA plate with a gap of 40 mm. As a control, a plate was only inoculated with *T. virens.* All plates were cultured at 25 °C in an incubator. Each experiment was repeated three times. After the mycelia of *T. virens* and *R. solani* formed an antagonistic line together, the mycelia were collected, shown as *R. solani* and *T. virens* ZT05 (Rs-Tv) and Tv.

Total RNAs were extracted from the mycelium tissue using a rapid plant RNA extraction kit Plant RNA Purification Reagent (Aidlab, Beijing, China) according the manufacturer’s instructions (Invitrogen). Concentration and purity of the extracted RNA were tested using a Nanodrop 2000 spectrophotometer. RNA integrity was checked through agarose gel electrophoresis, and the RNA integrity number (RIN) values were measured by an Agilent 2100 analyzer. The results satisfied the requirement that the total RNA be more than 5 g, the concentration more than 200 ng/L, and the OD 260/280 between 1.8 and 2.2 [61,62].

#### 2.4.2. Library Preparation and Illumina Sequencing

The cDNA was shotgun sequenced (101-bp paired-end reads) with a Illumina HiSeq 4000 instrument (Illumina, San Diego, CA, USA) using a customer sequencing service (Majorbio Co., Ltd., Shanghai, China). The sequencing reads were statistically analyzed, and quality assessed by FASTQC (http://www.bioinformatics.babraham.ac.uk/pro jects/fastqc/), and then processed by Trimmomatic to remove adapter sequences and low-quality reads with average quality scores lower than 15. Reads that were less than 50 base pairs (bp) after trimming were also excluded from further genome mapping [63,64].

The raw paired end reads were trimmed, and quality controlled by SeqPrep (https://github.com/jstjohn/SeqPrep) and Sickle (https://github.com/najoshi/sickle) with default parameters. Then, clean reads were separately aligned to reference genome with orientation mode using TopHat (http://tophat.cbcb.umd.edu/, version 2.0.0) [65] software. The mapping criteria of bowtie was as follows: sequencing reads should be uniquely matched to the genome allowing up to two mismatches, without insertions or deletions. Then, the region of gene was expanded following depths of sites and the operon was obtained. In addition, the whole genome was split into multiple 15 kbp windows that shared 5 kbp. New transcribed regions were defined as more than two consecutive windows without overlapped region of gene, where at least two reads were mapped per window in the same orientation. We compered the readings of the samples to the Trichoderma_virens (TRIVI v2.0, https://www.ncbi.nlm.nih.gov/genome/?term=txid29875[orgn]), reference genomic sequence using the TopHat software (http://tophat.cbcb.umd.edu/version2.0.0).

#### 2.4.3. Differential Expression Analysis and Function Gene Selection

To identify DEGs (differential expression genes) between two different samples, the expression level of each transcript was calculated according to the fragments per kilobase of exon per million mapped reads (FRKM) method. RSEM (http://deweylab.biostat.wisc.edu/rsem/) [66] was used to quantify gene abundances. R statistical package software EdgeR (Empirical analysis of Digital Gene Expression in R, http://www.bioconductor.org/packages/2.12/bioc/html/edgeR.html) [67] was utilized for differential expression analysis (|log2FC| ≥ 1, *p*-adjust < 0.05). In addition, functional enrichment analysis Swiss-Prot were performed to identify which DEGs were significantly enriched in Swiss-Prot terms.

### 2.5. Validation of Quantitative qRT-PCR of the Transcriptome Data

In order to verify the reliability of the transcriptome sequencing, eight upregulated genes were selected for PCR tests, including TRIVIDRAFT_56363, TRIVIDRAFT_178019, TRIVIDRAFT_213202, TRIVIDRAFT_72072, TRIVIDRAFT_226194, TRIVIDRAFT_63956, TRIVIDRAFT_61106, and TRIVIDRAFT_80583. Each gene was detected three times. The primers of the selected genes are described in Table 1. Total RNAs were extracted from the mycelium tissue using a rapid plant RNA extraction kit Plant RNA Purification Reagent (Aidlab, Beijing, China) according to the manufacturer’s instructions (Invitrogen, Carlsbad, CA, USA). The cDNA synthesis was performed following instructions provided in the RevertAid First Strand cDNA Synthesis Kit (Thermo Scientific, Waltham, MA, USA). Each reaction for the cDNA synthesis included 5× qRT SuperMix II 4 µL, the template RNA (total RNA): 1 µg, and RNase Free ddH2O 16 µL. Reactions were done at 50 °C 15 min, followed by 80 °C, for 2 min [68]. The Trans Start^®^ Top Green qPCR SuperMix Kit was used for the qRT-PCR reactions. Each reaction for the qPCR reaction included ChamQ SYBR Color qPCR Master Mix 2 × 10 µL, Primer F 5 µM 0.4 µL, Primer R 5 µM 0.4 µL, Template (DNA) 2 µL, and ddH2O 7.2 µL [69].

### 2.6. Data Analysis

Data were analyzed using Excel 2017. Origin 2019b (Origin Lab Corporation, MA, United States) was used for graph plotting. Fluorescence quantitative data were analyzed using the 2 – (∆∆Ct) method [70]. The sequencing data were deposited in the NCBI/SRA database (Bioproject: PRJNA526386; Sequence Read Archive Database under accession number SRR8706306–SRR8706311).

## 3. Results

### 3.1. Antagonistic Effect of T. virens ZT05 against R. solani

*T. virens* ZT05 showed a significant antagonistic effect against *R. solani* (Figure 1). After 48 h of side-by-side culture, the inhibitory rate (A) and relative inhibitory rate (B) of T. virens against *R. solani* increased from 12.99% to 42.82% and from 1.09% to 1.35%, respectively. After contacting *T. virens*, the inhibitory effect on the growth of *R. solani* was increased and reached max at 80 h of culture with the inhibitory rate of 42.82% and the relative inhibitory rate of 1.35%. *T. virens* ZT05 grew fast with a competition coefficient of II. They formed antagonistic lines with *R. solani* during the side-by-side culture and gradually covered the growth area of *R. solani*. The aerial mycelium of *R. solani* shrank and became sparse gradually until growth ceased (C). In addition, *T. virens* ZT05 had hyperparasitic ability against *R. solani*. The SEM results showed that the mycelia of *T. virens* ZT05 grew parallel to the mycelia of *R. solani*, and then spirally wrapped around or penetrated the pathogenic mycelia of *R. solani* (Figure 2).

### 3.2. Inhibitory Effect of the Metabolites of T. virens ZT05 on R. solani

#### 3.2.1. Inhibitory Effect of Nonvolatile Metabolites of *T. virens* ZT05 on *R. solani*

The nonvolatile metabolites of *T. virens* showed a significant inhibitory effect on *R. solani* (Figure 3). The inhibition rate increased gradually between 0–40 h and reached 63.32% at 40 h. The inhibition rate reached a plateau between 40–88 h with inhibited mycelium growth of *R. solani.* The mycelium appeared sparse and weak, with ceased growth.

#### 3.2.2. Inhibitory Effect of Nonvolatile Metabolites of *T. virens* ZT05 on Enzyme Activities of *R. solani*

The ethyl acetate extract of *T. virens* ZT05 significantly affected the expression of *R. solani* mycelium proteins, including CAT, SOD, POD, GSH-Px, MDA, and soluble proteins (Figure 4). In the *T. virens* ZT05-treated group, the enzyme activities of *R. solani* CAT, SOD, and POD started to increase at 3 h, reached peaks at 9 h, and then decreased. The maximum values of enzyme activities were 34.66, 45.07, and 8.55 for CAT, SOD, and POD in the treated group. Their enzyme activities started to decrease slowly after 24 h and reached minimum values at 48 h. The minimum values of enzyme activities for CAT, SOD, and POD were 478.02%, 26.02%, and 36.68% less than the control group. The *T. virens* ZT05-treated group showed decreased GSH-Px enzyme activity than that of the control group. The enzyme activity reached a maximum value of 1.78 at 9 h and gradually decreased until 48 h with a minimum value of 0.10. The *T. virens* ZT05-treated group showed increasing MDA enzyme activity until 48 h with a maximum value of 4.17. The *T. virens* ZT05-treated group showed decreasing content of soluble proteins until 48 h, with a minimum value of 0.17, which was 47.67% less than that of the control group.

#### 3.2.3. Inhibitory Effect of Volatile Metabolites of *T. virens* ZT05 on *R. solani*

The volatile metabolites of *T. virens* ZT05 showed an inhibitory effect of *R. solani* during the up-and-down culture (Figure 5). After five days of up-and-down culture (*T. virens* ZT05 on the bottom and *R. solani* on the top), the inhibitory effect on *R. solani* reached 80.10% with the slow and sparse growth of *R. solani*.

### 3.3. Transcriptome Sequencing Analysis of the Side-by-Side Cultured T. virens ZT05 and R. solani

#### 3.3.1. Analysis of Transcriptome Sequencing Data

The junction of *T. virens* ZT05 and *R. solani* after 48 h of side-by-side culture was sampled for Illumina sequencing. After filtering the data, a total of 91,307,884.00 clean reads were obtained, which accounted for 99% of the total sequencing sequence (Table 2). The GC content of each sample was between 52.66–54.17%. Q30 was greater than 95%. Their comparison rate to the reference genome was 60.12–92.91%. Hence, the sequencing quality was high and met the requirements of subsequent analysis.

#### 3.3.2. Screening of Hyperparasitic Functional Genes

Seven genes related to pathogen recognition and signal transduction were significantly upregulated during the plate paired culture (Figure 6), among which two genes were extracellular proteases, one gene belonged to oligopeptide transporters, and four genes were G-protein coupled receptors. Thirteen genes related to the hyperparasitic genes, among which, six chitinase genes were identified (Figure 7 and Appendix A), and Log_2_FC value of TRIVIDRAFT_178019 was the highest, shown as 2.62. In addition, six glucanase genes were identified, among which the Log_2_FC value of TRIVIDRAFT_72072 was the highest, shown as 2.02. We also screened one proteasome gene, and its Log_2_FC value was 1.0876.

#### 3.3.3. Screening of Antibiosis Functional Genes

The antibiosis function of *Trichoderma* plays an important role in the antagonistic mechanism. Studies have shown that *Trichoderma* species metabolized many chemicals and enzymes that were antagonistic. For example, 70 antifungal metabolites have been identified. These metabolites functioned antagonistically via inhibiting the growth of pathogenic fungi. Several antibiosis function genes were identified through the initial screening, nine reductase genes (Figure 8A), including two tetracycline resistance genes (Figure 8A), eight heat shock response genes (Figure 8B), two multidrug resistance transporter genes (Figure 8B), eight ABC efflux transporter genes (Figure 8C), and oxidative stress response gene (Figure 8C). A total of 30 genes related to antibiosis function were identified (Figure 8, Appendix A).

#### 3.3.4. qRT-PCR Quantitative Analysis

In the *Trichoderma* virens ZT05-treated group (Rs-Tv), several gene expressions were upregulated as compared with the control, consistent with the transcriptome sequencing results, including TRIVIDRAFT_56363, TRIVIDRAFT_178019, TRIVIDRAFT_213202, TRIVIDRAFT_72072, TRIVIDRAFT_226194, TRIVIDRAFT_63956, TRIVIDRAFT_61106, and TRIVIDRAFT_80583 (Figure 9).

## 4. Discussion

*Trichoderma* inhibits the growth and reproduction of pathogenic microorganisms mainly via competition, hyperparasitism, and antibiosis. Competition is a process that occurs when an organism lacks nutrients and living space. The “hunger” caused by nutrition deficiency is the most common cause of microbial death. The use of a biological control to compete for limited nutrition is an important way to prevent plant diseases [71]. In this study, *T. virens* ZT05 and *R. solani* formed an antagonistic line. With the increase in antagonistic time, the mycelium of *R. solani* gradually became thinner and stopped growing. The results of electron microscopy showed that the mycelium of *T. virens* ZT05 was spirally wrapped around and attached to the mycelium of *R. solani* during the growth phase, consistent with the results of Harwoko et al. [72], Mukherjee et al. [73], and Li et al. [74]. Hyperparasitism is the main mechanism of biological control of the *Trichoderma* species. *Trichoderma* recognizes pathogens by identifying the lectin secreted by the pathogens [75,76]. After recognizing the host pathogen, *Trichoderma* induces a series of hyper parasite-related signaling pathways in the body. Many genes encoding extracellular proteases, oligopeptide transporters, and GPCRs are expressed when the *Trichoderma* contacts the host pathogen or before the contact [77,78]. Most encoded proteins belong to the subtilisin-like serine protein family. Their coding genes are significantly upregulated during the recognition of host pathogen by *Trichoderma* [78]. During the hyperparasitic process, *Trichoderma* secretes cell wall degrading enzymes (CWDEs), including chitinases, cellulases, xylanases, glucanases, and proteinases [79]. CWDEs and secondary metabolites synergistically degrade the cell wall of the pathogen [80]. We found several genes to be upregulated, including seven genes related to recognition and signal transduction, six chitinase genes, six glucanase, and one protease gene. Ravindra et al. [81] showed that *T. virens* IMI 304061 had a significant antagonistic effect on *Sclerotium rolfsii*. The *pgy1* and *ecm33* genes of *T. virens* IMI 304061 were related to the hyperparasite ability. Knocking out *pgy1* and *ecm33* genes resulted in the loss of the ability of *T. virens* IMI 304061 to transmit signals and secrete metabolites. Dautt-Castro et al. [82] showed that *T. virens* Gv29–8 had a significant antagonistic effect on *R. solani*, Sclerotium rolfsii, and Fusarium oxysporum. The *tbrg-1* gene was associated with the hyperparasitic effect of *Trichoderma*. The gene spi encoded protease. The gene *chtl* encoded chitin. The gene *gliP* encoded secondary metabolites. Saravanakumar et al. [83] showed that genes associated with parasitism and secretion of secondary metabolites in *Trichoderma* were upregulated during antagonism of *T. virens* against *Macrophomina phaseolina* (MP), *Fusarium graminearum* (FG), and 104 *Botrytis cinerea* (BC). In addition, the secondary metabolites secreted by *Trichoderma* have a synergistic effect during the antagonistic process. Ngikoh et al. [84] showed that during culturing of *T. virens* UKM1 with OPEFB substrate, *Trichoderma* degraded OPEFB by secreting cellulase and glucanase. The contents of cellulase and glucanase were the highest on the seventh day of culture. Antibiosis is one of the important mechanisms for the biological control of *Trichoderma*. During the antagonistic process of *T. virens* against pathogens, *T. virens* produced volatile metabolites such as ethylene, hydrocyanic acid, ethanol, acetaldehyde, ketones, chloramphenicol, chloramphenicol, and nonvolatile metabolites such as trichome in, antibacterial peptides, chitinase, and β-1,3-glucanase [34,35]. During the interaction between *Trichoderma* and the host pathogen, the host secreted toxic substances to inhibit the growth of *Trichoderma*. The *Trichoderma* degrades the toxic substances through ABC transport proteins, heat shock proteins, etc., thereby alleviating the inhibition induced by the host [37,38]. The ABC transporter contains ATP binding sites and is the primary-secondary transport system of the cell. It transports the bound substrate to the outside of the cell membrane to avoid cell damage. It is directly related to the resistance of an organism to various exogenous substances [85,86]. Ruocco et al. [87] showed that the expression of *Taabc2* gene, encoding the ABC transporter, in *T. virens* was affected by the pathogenic secretions. Knocking out the *Taabc2* gene resulted in the loss of resistance to toxic substances secreted by *Pythium ultimum* and *R. solani* and the loss of biological control ability of *T. virens*. In this study, the nonvolatile metabolites of *T. virens* ZT05 significantly inhibited the growth of *R. solani*. The enzyme activities of CAT, SOD, POD, and GSH-Px and the expression of soluble proteins gradually decreased till minimum values at 48 h of culture. The volatile metabolites of *T. virens* ZT05 significantly inhibited the growth of *R. solani*. After five days of up-and-down culture, the size of *R. solani* decreased significantly. Several genes were identified during the transcriptome analysis of the side-by-side culture of *T. virens* ZT05 and *R. solani*, including two genes encoding tetracycline resistance proteins, nine genes encoding reductases, eight genes for the heat shock responses, one gene for the oxidative stress responses, eight genes encoding ABC efflux transporters, and two genes encoding multidrug resistance transporters. Bae et al. [36] extracted metabolite KACC (Korea Agricultural Culture Collection, 40557) from *T. atroviride* and metabolite KACC 40929 from *T. virens.* These two metabolites showed significant inhibitory effects on *P. litchii*. The expression of genes related to secondary metabolites was upregulated. Ruocco et al. [87] found that the gene *Taabc2* encoded the ATP-binding cassette transporter. The expression of *Taabc2* gene was upregulated when the level of environmental toxins increased around the mycelium of *Trichoderma*. Srivastava et al. [88] showed that the antagonistic effect of *Trichoderma virens* against *R. solani* and *Sclerotium rolfsii* resulted in the upregulation of genes *TgaA* and *TgaB*. Additionally, the *gene TvGST* was associated with the antibiosis of *T. virens*. Montero-Barrientos et al. [89] showed that the expression of *T. virens* T59 *hp23* gene was upregulated after temperature treatment and 4 h of 10 % ethanol immersion, which resulted in improved adaptability to extreme environments and resistance to toxic substances of *T. virens* T59.

## 5. Conclusions

*T. virens* ZT05 has strong competition for nutrients and living space, and hyperparasitism to *R. solani*. The volatile and nonvolatile metabolites of *Trichoderma* had significant inhibitory effects on *R. solani*. During the co-culture, the expression of biocontrol genes from *T. virens* ZT05 was significantly upregulated, including hyperparasitic functional genes and antibiosis functional genes.

## Figures and Tables

**Figure 1 plants-09-00912-f001:**
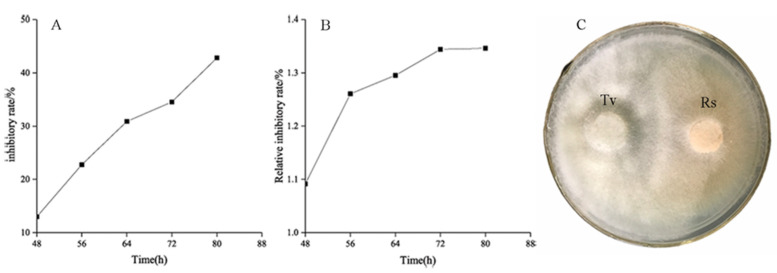
Antagonistic effect of *T. virens* ZT05 against *R. solani* ((**A**), inhibition rate of *T. virens* ZT05 against *R. solani*; (**B**), relative inhibition effect of *T. virens* ZT05 against *R. solani*; (**C**), dural culture of *T. virens* ZT05 and *R. solani*; Tv, *T. virens* ZT05; Rs, *R. solani*).

**Figure 2 plants-09-00912-f002:**
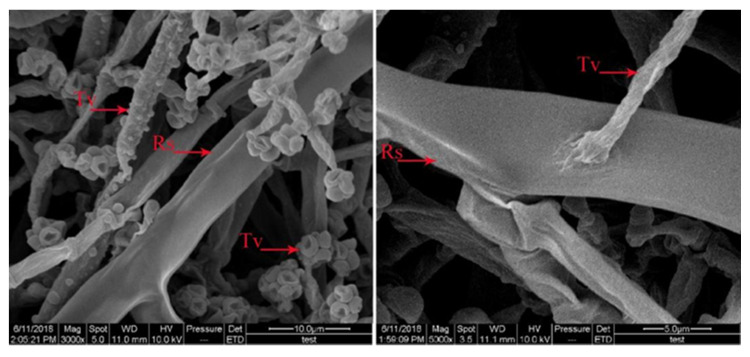
Electron microscopic observation of *T. virens* ZT05 and *R. solani* in dural culture (Tv, *T. virens* ZT05 and Rs, *R. solani*).

**Figure 3 plants-09-00912-f003:**
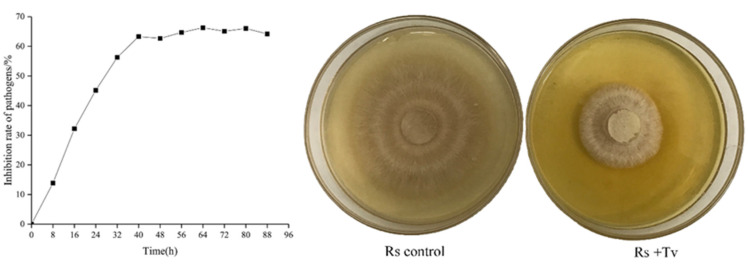
Inhibitory effect of nonvolatile metabolites of *T. virens* ZT05 on *R. solani* (Rs) (Control, 20% Tween 80 solution; Rs + Tv, 20% nonvolatile metabolites solution).

**Figure 4 plants-09-00912-f004:**
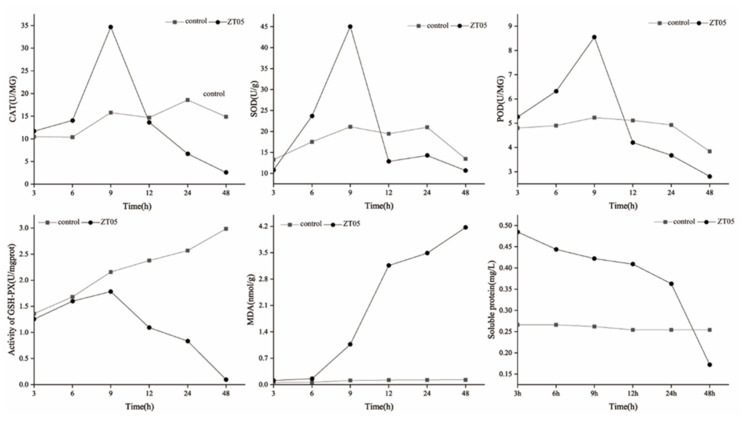
Effect of nonvolatile metabolites on the enzyme activity of *R. solani* (Control, *R. Solani* mycelium soaking in 20% Tween 80 solution and ZT05, *R. Solani* mycelium soaking in 20% nonvolatile metabolites solution of *T. virens* ZT05).

**Figure 5 plants-09-00912-f005:**
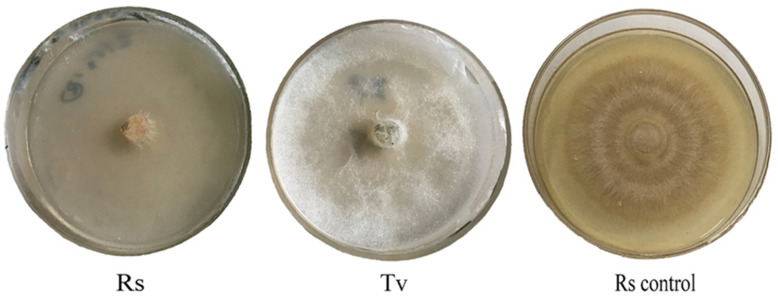
Inhibitory effect of volatile metabolites of *T. virens* ZT05 on *R. solani* (Rs, *R. solani*; Tv, *T. virens* ZT05; and Rs control, *R. solani* control.).

**Figure 6 plants-09-00912-f006:**
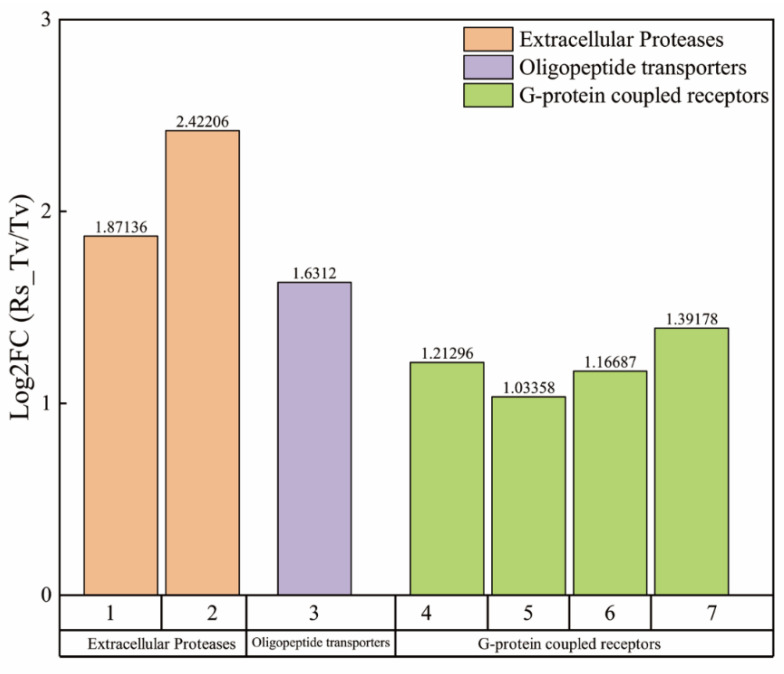
Genes related to recognition and signal transduction in mycoparasitic process of *T. virens* ZT05. 1, 2, 3, 4, 5, 6, and 7 represent TRIVIDRAFT_81735, TRIVIDRAFT_63956, TRIVIDRAFT_58191, TRIVIDRAFT_35938, TRIVIDRAFT_30459, TRIVIDRAFT_59177, TRIVIDRAFT_138511, respectively.

**Figure 7 plants-09-00912-f007:**
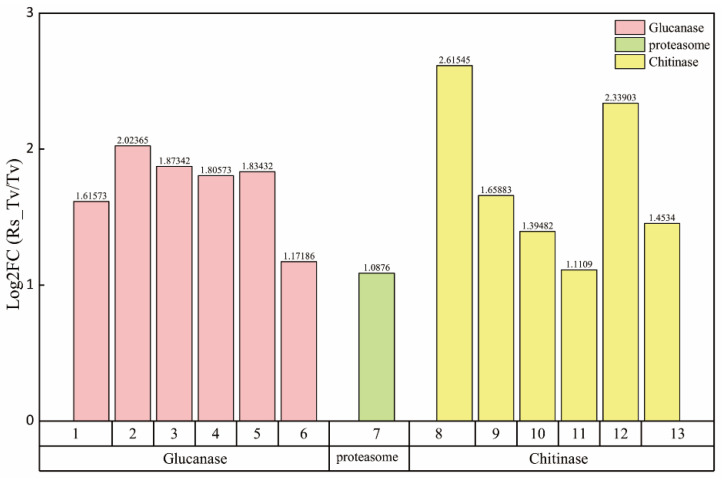
Chitinase, glucanase, and protease genes from T. virens ZT05. 1, 2, 3, 4, 5, 6, 7, 8, 9, 10, 11, 12, and 13 represent TRIVIDRAFT_89797, TRIVIDRAFT_72072, TRIVIDRAFT_27891, TRIVIDRAFT_42536, TRIVIDRAFT_28149, TRIVIDRAFT_76895, TRIVIDRAFT_176639, TRIVIDRAFT_178019, TRIVIDRAFT_89999, ECH1, TRIVIDRAFT_69839, TRIVIDRAFT_213202, CHT1.1 gene name, respectively.

**Figure 8 plants-09-00912-f008:**
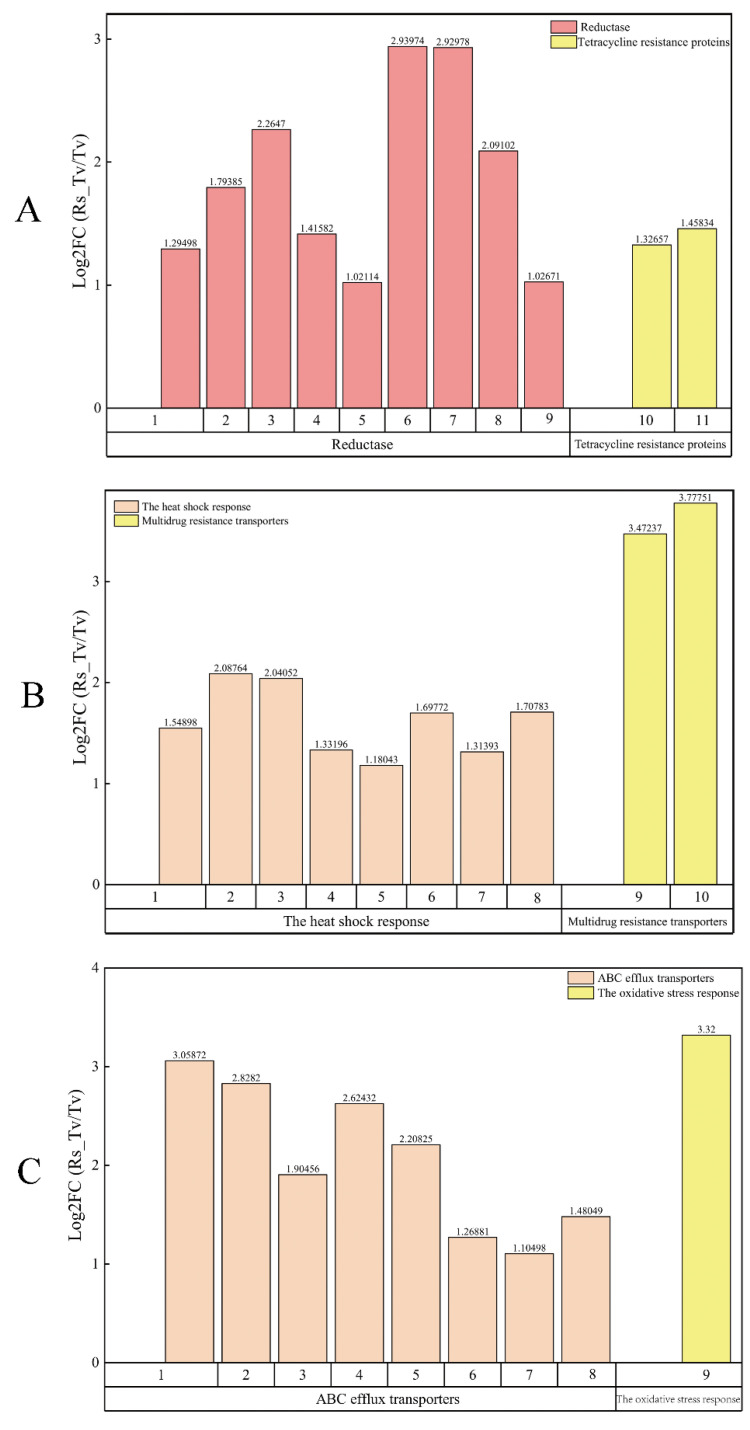
Resistance protein genes from *T. virens* ZT05. (**A**): 1, 2, 3, 4, 5, 6, 7, 8, 9, 10, and 11 represent TRIVIDRAFT_62654, TRIVIDRAFT_211837, TRIVIDRAFT_38645, TRIVIDRAFT_42391, TRIVIDRAFT_45041, TRIVIDRAFT_50977, TRIVIDRAFT_68923, TRIVIDRAFT_69465, TRIVIDRAFT_71556, TRIVIDRAFT_57595, TRIVIDRAFT_219995 gene name respectively. (**B**): 1,2,3,4,5,6,7,8,9,10 represent TRIVIDRAFT_216898, TRIVIDRAFT_80583, TRIVIDRAFT_215292, TRIVIDRAFT_210885, TRIVIDRAFT_89650, TRIVIDRAFT_195722, TRIVIDRAFT_78895, TRIVIDRAFT_217094, TRIVIDRAFT_76205, TRIVIDRAFT_192676 gene names, respectively. (**C**): TRIVIDRAFT_33722, TRIVIDRAFT_52608, TRIVIDRAFT_86623, TRIVIDRAFT_36031, TRIVIDRAFT_85589, TRIVIDRAFT_190418, TRIVIDRAFT_45576, TRIVIDRAFT_83793, TRIVIDRAFT_207997 gene names, respectively.

**Figure 9 plants-09-00912-f009:**
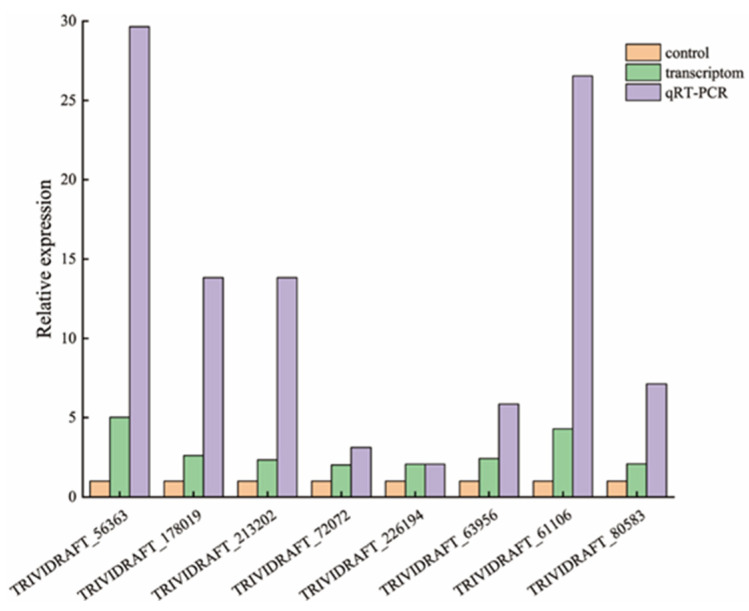
qRT-PCR quantitative analysis.

**Table 1 plants-09-00912-t001:** qRT-PCR primer sequences.

Gene	Primer Name	Sequence (5′–3′)
TRIVIDRAFT_56363	1-TRIVIDRAFT_56363-F 1-TRIVIDRAFT_56363-R	TATATGAGCGCTGCTAAGAT AAATTGGGACTTGTGAGTGT
TRIVIDRAFT_178019	2-TRIVIDRAFT_178019-3F 2-TRIVIDRAFT_178019-3R	GGATCCAGATTCAGTTCTAT GGATCCAGATTCAGTTCTAT
TRIVIDRAFT_213202	3-TRIVIDRAFT_213202-3F 3-TRIVIDRAFT_213202-3R	GGATCCAGATTCAGTTCTAT GGATCCAGATTCAGTTCTAT
TRIVIDRAFT_72072	4-TRIVIDRAFT_72072-F 4-TRIVIDRAFT_72072-R	GGATCCAGATTCAGTTCTAT GGATCCAGATTCAGTTCTAT
TRIVIDRAFT_226194	5-TRIVIDRAFT_226194-2F 5-TRIVIDRAFT_226194-2R	GGATCCAGATTCAGTTCTAT CAGAAATCTGCATTTGCAAG
TRIVIDRAFT_63956	6-TRIVIDRAFT_63956-F 6-TRIVIDRAFT_63956-R	CAGAAATCTGCATTTGCAAG CGTGCTTGCGATGTGTAAGT
TRIVIDRAFT_61106	7-TRIVIDRAFT_61106-F 7-TRIVIDRAFT_61106-R	CTCGGACAACAGCCAGTTTC CTCGGACAACAGCCAGTTTC
TRIVIDRAFT_80583	8-TRIVIDRAFT_80583-F 8-TRIVIDRAFT_80583-R	CTCGGACAACAGCCAGTTTC CTCGGACAACAGCCAGTTTC

**Table 2 plants-09-00912-t002:** Results of transcriptome sequencing.

Sample	Raw Reads	Clean Reads	Error Rate (%)	Q30 (%)	GC Content (%)	Total Mapped
Tv	47,012,424.00	46,570,488.67 (99.00%)	0.02	95.91	54.17	43,264,365.67 (92.91%)
Rs-Tv	45,402,895.33	44,737,395.33 (99.00%)	0.02	95.50	52.66	26,959,925.00 (60.12%)
Total	92,415,319.33	91,307,884.00 (99.00%)	0.02	95.71	53.42	70,224,291.00 (76.52%)

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
