# Peer review of "Inhibitory Mechanism of Trichoderma virens ZT05 on Rhizoctonia solani"

_plants, 2020, doi:10.3390/plants9070912_

Round 1

Reviewer 1 Report

This is an interesting article about the inhibitory mechanism of Trichoderma virens on Rhizoctonia solani through the side-by-side culture.

The paper is well written, with many topical references, but in my opinion, the Discussion section can be improved with very recent results in the field, because there is a lot of data in the literature on this subject. This is not a new topic at all. I suggest you to explain and clarify obviously the cause that this topic has been selected for the current study namely what is the level of novelty, compared to the other cited results.

There are also some minor observations:

Lines 21, 108, 109, 111, etc. Please italicize the scientific names T. virens and R. solani and check carefully in the whole document.

Lines 286-292. Please reformulate this paragraph, because is confusing.

Line 383. Please improve the conclusions section by clearly mentioning the importance of the results obtained for research in the field.

Reviewer 2 Report

The work is important for the research field and is being well conducted. I have only small suggestions:

Tables referring to gene expression can be attached. They are polluted of data. Instead, I would add graphics with the desired categories.
All RNA data must be made available in public databases and the access number added in the article
The cutoff point used for differential expression no longer applies. With this restriction, you lose a lot of information about genes that are poorly expressed, such as cell signaling genes, and that are important. So I suggest putting all the genes statistically different. In addition, they are also important for building the gene expression network.
Add a picture of the gene expression network, it will add a lot of information to the discussion.
Finally, it is no longer used to validate RNAseq correlating with RT-qPCR. They are techniques with different sensitivity and if the RNAseq is well conducted, which is the case with this work, it is sufficient to trust the data.
